# Community based hematological reference intervals among apparently healthy adolescents aged 12-17 years in Mekelle city, Tigrai, northern Ethiopia: A cross sectional study

Hagos Haileslasie [1]*, Aster Tsegaye[2], Gebreyohanes Teklehaymanot [3], Getachew Belay[1], Gebreslassie Gebremariam [3], Gebremedhin Gebremichail[1], Brhane Tesfanchal[1], Kelali Kaleaye[4], Lemlem Legesse[4], Gebre Adhanom[1], Fitsum Mardu [1], Aderajew Gebrewahd[1], Gebrehiwet Tesfay[1], Ataklti Gebertsadik[4]

1 Department of Medical Laboratory Sciences, College of Medicine and Health Sciences, Adigrat University, Adigrat, Tigrai, Ethiopia, 2 Department of Medical Laboratory Sciences, College of Health Sciences, Addis Ababa University, Addis Ababa, Ethiopia, 3 Department of Medical Laboratory Sciences, College of Medicine and Health Sciences, Mekelle University, Mekelle, Tigrai, Ethiopia, 4 Laboratory Diagnostic, Research and Quality Assurance Directorate, Tigrai Health Research Institute, Mekelle, Tigrai, Ethiopia

* hagoshaileslasie78@gmail.com

## Abstract

### Background

Hematological reference intervals are important in clinical and diagnostic management for the assessment of health and disease conditions. Hematological reference intervals are better to be established based on gender and age differences as these are among the main affecting factors.

### Objective

The aim of this study was to establish hematological reference intervals among apparently healthy adolescents aged 12–17 years in Mekelle City, Tigrai, Northern Ethiopia, 2019.

### Method

A community-based cross-sectional study was conducted in 249 adolescents aged 12–17 years from December 2018 to May 2019. About 4ml of blood sample was collected from each study participant using vacutainer tube containing $K_2EDTA$. Hematological parameters were analyzed using Sysmex KX-21N hematology analyzer (Sysmex Corporation Kobe, Japan). Data were entered and analyzed using SPSS version 23. Both parametric and non-parametric analyses were used to calculate the median and 95% of reference intervals. The 97.5th and 2.5th percentiles were calculated using descriptive statistics for the upper and lower reference limits of the study participants. Differences in reference intervals between male and female participants were evaluated using the Mann–Whitney U test.

**Data Availability Statement:** All relevant data are within the paper and its Supporting Information files.

**Funding:** The author(s) received no specific funding for this work.

**Competing interests:** The authors have declared that no competing interests exist.

## Result

Among the 249 participants 122 (49%) were males and 127 (51%) were females with the median age of 14.5 (range 12 to 17) years were recruited in this study. The median and the 95% reference intervals of hematological parameters were determined. The 95% RIs were: Red blood cells ($10^{12}$/Liter) 4.6–5.9 (Males) and 4.3–5.6 (Females), White blood cells ($10^9$/Liter) 2.9–9.6 (Males) and 3.4–10.2 (Females), Hemoglobin (g/dl) 12.6–17.1 (Males) and 12–15.4 (Females), Platelets ($10^9$/Liter) 138–364 (Males) and 151–462 (Females). Almost all of the hematological parameters showed significant differences ($p<0.05$) across gender.

## Conclusion

The hematological reference intervals established in this study showed a difference based on gender. We suggest preparing and using distinct local reference intervals for males and females separately.

## Background

Reference interval (RI) is a standard component of reporting laboratory results and important to transform a numerical value into clinically meaningful information. It is intended to inform the clinical care provider that laboratory values within the interval indicate a non-diseased condition. The most common approach is to base RI on the central 95% of laboratory test values observed for a reference population that is free of disease that influence the laboratory test result, as many diseases are asymptomatic [1].

Human beings are characterized by a dynamic period of growth and development in their lives. The different periods of life possess special hematological development. The chemical makeup of the circulating red blood cells and fetal hemoglobin content in the first days and months is not identical to in later life [2]. In old age, there is increased replacement with fatty marrow [3]. This may affect the hematological parameters and result in different reference intervals for the difference among age groups [4] and gender [5]. Thus, knowledge of the reference values during the dynamic period of growth and development in an individual is important for correct interpretation of a disease condition.

Hematological reference intervals are important in clinical practice for the assessment of health and disease, underscoring the importance of establishing population-appropriate values. Such parameters are important for measuring disease progression, response to therapy, and in the assessment of adverse reactions to therapy [6]. Due to the prevailing disease epidemics in sub-Saharan Africa, the normal hematological indices for these populations become critical in supporting decisions concerning treatment initiation and disease management [7]. Establishment of well-controlled, reliable RI is an important mission for all clinical laboratories. The clinical laboratory standards institute (CLSI) [6, 8] guideline recommends each laboratory to establish its own RIs from the local population or validate them if derived from a different setting. RIs need to be verified periodically every 5 years, to capture changes in the community over time [8].

Most reference values of hematological tests currently being used in Africa are derived the data collected from populations living in developed countries [9]. The RIs adopted from people living in developed countries differ from the people living in sub-Saharan countries which are endemic to different diseases [10]. Most blood specimens of Africans have been found to

have lower Hemoglobin (Hgb), Hematocrit (Hct), Red blood cell count (RBC), mean corpuscular volume (MCV), neutrophil and platelet counts compared to the developed countries [11].

The hematological reference intervals which are used for clinical trials in sub-Saharan Africa follow the WHO reference standard guidelines which are different from the Ethiopian guidelines. However, typical Laboratory parameters in different communities may vary based on race, age, gender, diet, local disease patterns and environmental characteristics [12]. Here, we present our efforts to generate RIs in adolescents separately for males and females for the local community.

## Methods

### Study area

A community based cross-sectional study was conducted from December 2018 to May 2019 in Mekelle City, Tigrai, and northern Ethiopia. Mekelle is the capital city of Tigrai regional state. It is located 780 kilometers north of Addis Ababa (the capital city of Ethiopia) at an altitude of 2,254 meters above sea level characterized by temperate climate [13]. Based on the 2007 Census conducted by the Central Statistical Agency) of Ethiopia (CSA), Mekelle has a total population of 215,914 (104,925 men and 110,989 women) [13, 14].

### Eligibility criteria

The apparently healthy individuals aged 12–17 years and lived at least for 5 years in the study area were included. Individuals with any chronic and acute illnesses, taking antibiotic treatment, recent history of blood loss, blood transfusion in the last one year, immunization in the last 6 months, major surgical procedures in the past 6 months, who have any intestinal and hemoparasites were excluded during data analysis.

### Sample size determination

The CLSI guideline, which was developed through the clinical and laboratory standards institute consensus processes, was employed to determine the sample size. These guidelines recommended that the best means to establish a reference interval is to collect samples from a sufficient number of reference individuals to yield a minimum of 120 samples for analysis, by non-parametric means for each partition (e.g. Gender, age range) [6, 8]. Therefore starting from the age of 12 years, males and females must separate for the establishment of reference intervals (thus, 240 participants are needed for both sexes). According to previous studies in other African countries, in large scale studies about 30% of apparently healthy individuals [7] do not qualify for RI determination for various reasons [7]. Considering a 30% withdrawal rate from data analysis, to reach the recommended sample size of 240, 344 individuals were enrolled (i.e. assuming 30% of 344 around 104 of the participants may withdraw during data collection).

### Reference population selection technique

Three sub-cities (Ayder, Hawelti and Semen) were selected from the total of seven sub-cities through random sampling techniques then the total sample (344) was categorized based on the relative house hold size in each sub-city using Probability Proportional to Size (PPS). The 3 sub-cities have a total of 68477 households (18266, 33319 and 16892 in Semen, Hawelti and Ayder respectively).The total study participants were proportionally distributed to each sub-cities based on their number of households. Accordingly, 92 from Semen, 167 from Hawelti

and 85 from Ayder adolescents were recruited. In each sub-city, there were 5 Kebelles (kebelle-is part of a sub city which is a small administrating part of the city), and then the numbers were distributed to recruit participants from each Kebelles based on the number of house hold. Finally the study participants were selected using the systematic sampling techniques ($K^{th}$ = 199). If the $K^{th}$ ($199^{th}$) house hold is not fulfilled the eligible criteria or does not have adolescent aged participant they were passed to the nearby household either to the next or former which fulfilled the criteria. Once volunteering participants fulfilling the eligibility criteria were identified by the health extension workers, they were invited to go to nearby health facilities for sample collection.

## Data collection

From the study participants who give assent and family consent, demographic data and a brief medical history were collected. The physical examination was performed by physicians. Blood specimens were collected for analyzing hematological parameters and hemoparasites. Stool specimens were collected for intestinal parasites examination and urine samples for urinalysis. The Laboratory results were shown to the participants upon their request through the health extension workers. But when their results show abnormal findings, they were linked to nearby health institutes for management and treatment according to the facilities guidelines. Socio-demographic and clinical data were collected using a structured questionnaire by translating to the local language. Data were collected by trained data collectors, physical examinations and anthropometric measurements were carried out by physicians.

## Sample collection and laboratory analysis

About 4ml of venous blood sample was collected from each participant using $K_2EDTA$ anticoagulated test tubes using multisampling needle from 8 am to 11 am. During the blood collection time participants were seated on a comfortable chair and tourniquet was applied for less than one minute until the vein is visible and anchored with the syringe. One's the vein is visible and anchored tourniquet was removed. All samples were labeled with unique identification number. The hematological samples were transported within 2-8°c cooled icebox and analyzed within 5 hours of collection. Before analysis the 2-8°c preserved sample was put in the room temperature for 30 minutes. Whole blood samples were used for analyzing hematological tests, blood morphology and hemoparasites identification. Complete blood counts were analyzed using Sysmex KX-21N an automated 3-part differential hematology analyzer (Sysmex Corporation Kobe, Japan). Both thick and thin blood film was performed for the detection and identification of hemoparasites as well as for blood morphology using giemsa staining methods. Even if the sample was collected from the apparently healthy participants by following stringent criteria and family folder of the health extension workers before sample collection but after sample collection left over plasma was screened for HBsAg, HCV, HIV and RPR in Tigrai blood bank using an ELISA technique.

Leak proof clean containers were used to collect urine and stool samples. Urinalysis was performed using both the reagent strip and microscopic analysis methods. Stool examination was performed using wet mount, Formol Ether sedimentation, kato-katz and modified acid fast staining techniques. Wet mount was performed on the collection sites immediately within 10 minutes after delivering the sample. But Formol Ether sedimentation, kato-katz and modified acid fast staining techniques was performed at Tigrai health research institution after collecting the necessary samples.

## Data quality control

The questionnaire was pre-tested with 5% (18) of individuals other than the study subjects. This pre-testing of a research instrument was entailed a critical examination of each question as to its clarity, understanding, wording, and meaning as understood by potential respondents to remove possible problems with the questions. Besides, adequate training was given to the data collectors before the data collection period. Participants were also being adequately oriented on how to collect specimens. The quality of laboratory analysis was maintained by following standard operating procedures of the pre analytical, analytical and post analytical stages. The Hospital was on the process of accreditation. Its maintenances protocol was controlled according to its standard procedures continuously. Accuracy was done by using the daily QC tests (Low, Normal and High values) and also reproducibility was checked. Between run and within run also checks to avoid carry over between the former and current samples.

## Data analysis

Data were cleaned, entered and analyzed using SPSS version 23. Both parametric and non-parametric analyses were performed since the data was not normally distributed when checked its normality. Data lower than first quartile ($Q1-1.5 \times IQR$), or higher than third quartile ($Q3 +1.5 \times IQR$) [15] were considered as outliers and the outliers was excluded. The median and 95th percentile reference intervals were determined by using 2.5th and 97.5th percentiles of each hematological parameter with descriptive statistics based on gender. When comparing the data of the three sub-cities and also the kebelle levels there was no significant difference in the RIs. Differences between males and females were evaluated using the Mann–Whitney U test. "P-value $< 0.05$" was considered as clinically significant at 95% confidence intervals.

## Ethical consideration

Ethical clearance was obtained from the research and ethical review committee of Department of Medical Laboratory Sciences of Addis Ababa University, Ethiopia. Before starting the study, permission was obtained from Tigrai regional health bureau and the selected sub-cities. Also, after explaining the purpose and relevance of the study, written consent was obtained from each guardian of study participant and assent from the participants before data collection. Confidentiality of information (results) was kept between the study participant and the investigators. All participants who were diagnosed positive for intestinal parasites were linked to nearby health institutions for treatment immediately. Cooperation letter was obtained from Wukro General Hospital to do the hematological tests.

# Results

## Socio demographic characteristics

In this study, 344 adolescents were participated and 249 study participants were eligible for final analysis. Out of the 249 eligible participants, 127 (51%) were females). The median age of study participants was 14.5 (SD 1.09) years. The overall exclusion rate was 27.6%. Out of the total samples collected, 50 (14.5%) were excluded due to the presence of intestinal parasites (ova of *S.mansoni* (33), ova of *H.nana* (12) and ova of *E.vermicularis* (5)), and 45(13.1%) were excluded due to incomplete information (15), due to outliers (20) and due to the presence of hemolysis (10).

## Hematological parameters of the study participants

The median values of RBCs, Hgb, HCT, absolute and percentile of mixed values (basophiles, Monocytes and Eosinophils) of the WBC differential and MCHC in females were lower than

males (p<0.05). Statistically significant higher values in MCV, platelets, RDW-CV, absolute Lymphocyte and Neutrophil counts were found in females (p<0.05). The Median and 95% RIs were as follows: **RBC** ($10^{12}$/L) 5.2, 4.6–5.9; **Hgb** (g/dl) 14.8,12.6–17.1; **HCT** (%) 44.8, 40–55; **MCHC** (g/dl) 32.5, 30–35.8; **WBC** ($10^9$/L) 5.4, 2.9–9.6; **Platelet** ($10^9$/L) 261, 138–364 for males; **RBC** ($10^{12}$/L) 4.9, 4.28–5.61; **Hgb** (g/dl) 14,12–15.4; **HCT** (%) 43.3, 38–47; **MCHC** (g/dl) 32.2, 30.4–34.2; **WBC** ($10^9$/L) 5.9, 3.4–10.2; **Plt** ($10^9$/L) 288,151–448 for females. There is no significant difference between males and females on the MCH, absolute Neutrophil count, percentile of lymphocytes and MPV values Table 1. When p value is <0.05 among the gender there is a significance difference (clinically difference) between Males and Females needs a separate reference interval based on the median value.

Comparing the upper and lower limits of our study to the currently in use reference intervals (company values) (Table 2) shows a higher proportions of out of range values observed for

**Table 1. Median and 95% RI of hematological parameters of the study participants in Mekelle, Tigrai, Northern Ethiopia, 2019 (n = 249).**

| Parameter | Gender | N | Median | RI (95%) | 2.5th centile (90% CI) | 97.5th centile (90% CI) | p-value |
|---|---|---|---|---|---|---|---|
| WBC($10^9$/L) | M | 122 | 5.4 | 2.9–9.6 | 2.6–3.3 | 9–9.7 | 0.027* |
| | F | 127 | 5.9 | 3.4–10.2 | 2.7–3.6 | 9.2–10.7 | |
| RBC($10^{12}$/L) | M | 122 | 5.2 | 4.59–5.91 | 4.45–4.63 | 5.88–5.98 | <0.001* |
| | F | 127 | 4.9 | 4.28–5.61 | 4.13–4.42 | 5.4–5.76 | |
| Hgb(g/dl) | M | 122 | 14.8 | 12.6–17.1 | 12.1–13 | 17–18.6 | <0.001* |
| | F | 127 | 14 | 12–15.4 | 11.4–12.4 | 15–16 | |
| HCT (%) | M | 122 | 44.8 | 40–55 | 38–40.8 | 51–58 | <0.00* |
| | F | 127 | 43.3 | 38–47 | 36–39 | 46–50 | |
| MCV(fl) | M | 122 | 86.7 | 76–94 | 74–80 | 93–95 | 0.019* |
| | F | 127 | 88 | 80–98 | 73–82 | 94–99 | |
| MCH(pg) | C | 249 | 28.3 | 24–31 | 23–24.8 | 30.8–31.8 | 0.085 |
| MCHC(g/dl) | M | 122 | 32.5 | 30–35.8 | 29.7–30.8 | 34.3–41.3 | 0.027* |
| | F | 127 | 32.2 | 30.4–34.2 | 29–30.9 | 33.8–34.3 | |
| PLT($10^9$/L) | M | 122 | 261 | 138–364 | 133–188 | 353–391 | <0.001* |
| | F | 127 | 288 | 151–448 | 140–181 | 413–467 | |
| RDW-CV (%) | M | 122 | 14.1 | 13–16 | 12.8–13.2 | 16–16.6 | 0.002* |
| | F | 127 | 13.7 | 12.5–15.6 | 12.4–12.8 | 15–17.6 | |
| Neutrophil($10^9$/L) | C | 249 | 3 | 0.9–6.8 | 0.8–1 | 6.3–7.2 | 0.113 |
| Mixed value($10^9$/L) | M | 122 | 0.8 | 0.3–3.8 | 0.2–0.4 | 2.- 4.2 | <0.001* |
| | F | 127 | 0.6 | 0.2–1.6 | 0.1–0.3 | 1.2–2.4 | |
| Lymphocyte($10^9$/L) | M | 122 | 2 | 1.2–3.3 | 0.9–1.3 | 3.1–3.7 | 0.004* |
| | F | 127 | 2.3 | 1.1–3.7 | 1–1.3 | 3.5–3.8 | |
| Neutrophil (%) | M | 122 | 45.5 | 23.7–64.7 | 21–24.9 | 62–74.5 | 0.036* |
| | F | 127 | 50.6 | 24.8–71 | 22–28.5 | 66–89.5 | |
| Mixed value (%) | M | 122 | 14.5 | 7.2–36.8 | 5.5–9.4 | 29–43.5 | <0.001* |
| | F | 127 | 10.5 | 4.8–22.7 | 1–5.1 | 19–37.7 | |
| Lymphocyte (%) | C | 249 | 39 | 19.8–61.5 | 17–23 | 59–64 | 0.48 |
| MPV(fl) | C | 249 | 10.6 | 8.8–13.1 | 8.6–8.9 | (12.6–13.6) | 0.567 |

**WBC**: White Blood Cell; **RBC**: Red Blood Cell; **Hgb**: Hemoglobin; **Hct**: Hematocrit; **MCV**: Mean Corpuscular Volume; **MCH**: Mean Corpuscular Hemoglobin; **MCHC**: Mean Corpuscular Hemoglobin Concentration; **PLT**: Platelet; **MPV**: Platelet Mean Volume **M**: Male; **F**: Female; **C**: Combined **N**: Number of participants.
**Mixed values**: are values which includes Basophiles, Monocytes & Eosinophiles displaying as one value as the machine is three differential *P < 0.05 by (Mann–Whitney U test) for comparison of medians between males and females.

**Table 2. Comparison of out of range values between new and old reference intervals for hematological parameters in Mekelle city, Tigrai, Northern Ethiopia, 2019 (n = 249).**

| Parameter | Gender | Current value | Instruments value | Lower range Frequency (%) | Upper range Frequency (%) | Total out of range Frequency (%) |
|---|---|---|---|---|---|---|
| WBC(10⁹/L) | M | 2.9–9.6 | | 31(25) | 2(1) | 33(27) |
| | F | 3.4–10.2 | 4.5–13 | 20(16) | 3(2) | 23(18) |
| RBC(10¹²/L) | M | 4.6–5.9 | 4.2–5.6 | 2(1) | 17(14) | 19(15) |
| | F | 4.3–5.6 | 4.1–5.3 | 3(2) | 9(7) | 12(9) |
| Hgb(g/dl) | M | 12.6–17.1 | 12.5–16.1 | 2(1) | 28(23) | 30(24) |
| | F | 12–15.4 | Dec-15 | 0 | 4(3) | 4(3) |
| HCT (%) | M | 40–55 | 36–47 | 5(4) | 28(23) | 33(27) |
| | F | 38–47 | 35–45 | 3(2) | 22(17) | 25(19) |
| MCV(fl) | M | 76–94 | 78–95 | 1(0.8) | 1(0.8) | 2(1) |
| | F | 80–98 | 78–95 | 2(1) | 0 | 2(1) |
| MCH(pg) | C | 24–31 | 26–32 | 13(5) | 2(0.8) | 15(6) |
| MCHC(g/dl) | M | 30–36 | 32–36 | 31(25) | 0 | 31(25.4) |
| | F | 30.4–34 | | 46(36) | 3(2) | 49(38) |
| PLT(10⁹/L) | M | 138–364 | 140–385 | 0 | 1(0.8) | 1(0.8) |
| | F | 151–462 | | 2(1.6) | 11(8) | 13(2.4) |
| Neutrophil(x10⁹/L) | C | 0.9–6.8 | 02-Jul | 68(27) | 1(0.4) | 69(27) |
| Mixed value(10⁹/L) | M | 0.3–3.8 | | 0 | 7(2.8) | 7(2.8) |
| | F | 0.2–1.6 | 0.24–1.6 | 0 | 0 | 0(0) |
| Lymphocyte(10⁹/L) | M | 1.2–3.3 | 01-Mar | 0 | 3(2.4) | 3(2.4) |
| | F | 1.1–3.7 | | 0 | 16(12.6) | 16(12.6) |
| Neutrophil (%) | M | 24–65 | 40–80 | 37(30) | 2(1.6) | 39(31) |
| | F | 24.78–71 | | 26(20) | 2(1.6) | 28(21) |
| Mixed value (%) | M | Jul-37 | Apr-18 | 2(1.6) | 33(27) | 35(28) |
| | F | May-23 | | 2(1.6) | 6(4.7) | 8(6) |
| Lymphocyte (%) | C | 20–62 | 20–40 | 0 | 104(42) | 104(42) |
| MPV(fl) | M | | 7.2–10.4 | | | |
| | F | | 7.5–11.5 | | | |
| | C | 8.8–13.1 | | 4(1.6) | 91(36.5) | 95(38.5) |

**WBC**: White Blood Cell; **RBC**: Red Blood Cell; **Hgb**: Hemoglobin; **Hct**: Hematocrit; **MCV**: Mean Corpuscular Volume; **MCH**: Mean Corpuscular Hemoglobin; **MCHC**: Mean Corpuscular Hemoglobin Concentration; **PLT**: Platelet; **MPV**: Platelet Mean Volume **M**: Male; **F**: Female; **C**: Combined **N**: Number of participants. **Mixed values**: are values which includes Basophiles, Monocytes & Eosinophiles displaying as one value as the machine is three differential.

RBCs 19(15%), WBC 33(27%), MCHC 31(25.4%), Hgb 30(24%), Hct 33(24%), Mixed value 35 (28%) and percentile neutrophils 39(31%) in males. In females, a higher proportion with out of range values were observed for RBCs 12(9%), WBC 23(18%), Hct 25 (19%), MCHC 49 (38%) and percentile neutrophils 28(21%). The combined value out of range when comparing with currently in use reference intervals (company ranges) were Lymphocyte percentile 104 (42%) MPV 95(38.5%) and absolute neutrophils 69(27%). The greatest proportion of the study participants with values outside the lower reference limits of reference intervals were observed for absolute neutrophil 68 (27%) while Lymphocyte percentile 104 (42%) had the greatest proportion of participants with values above the upper reference limits of the currently in use reference intervals. When comparing between our study reference intervals and the currently in use reference interval for WBC test parameters, 51(20%) out of the 249 participants were considered leucopenia while 5(2%) were considered as leukocytosis.

**Table 3. WBC, Hgb, Hct, MCV, Plt and WBC subsets for apparently healthy Mekelle city children compared to previously published Tanzanian, Ugandan, Zimbabwe and industrialized country reference intervals.**

| Parameter | Gender | Current Study (95% RI) | Instruments Values [16] | South west Ethiopia [17] | Tanzanian (95% RI) [18] | Uganda (95% RI) [19] | United States/Europe(95% RI) [20] | Zimbabwe (95% RI) [21] |
|---|---|---|---|---|---|---|---|---|
| WBC(10$^9$/L) | M | 2.9–9.6 | | 4.0–11.7 | | | | 3.25–8.64 |
| | F | 3.4–10.2 | | 3.7–11.4 | | | | 3.3–9.8 |
| | C | | 4.5–13 | | 3.2–10.3 | 4.1–10.7 | 4.5–13 | |
| RBC(10$^{12}$/L) | M | 4.6–5.9 | 4.2–5.6 | 4.06–6.57 | - | - | - | 4.47–6.47 |
| | F | 4.3–5.6 | 4.1–4.5 | 4.32–5.63 | | | | 4.8–5.83 |
| Hgb(g/dl) | M | 12.6–17.1 | 12.5–16.1 | 12–19.6 | 10.8–17 | 11.2–15.9 | 13–16 | 12.1–17.4 |
| | F | 12–15.4 | 12–15.4 | 11.6–15.9 | 10–14.9 | 9.9–14.5 | Dec-16 | 11.1–15.7 |
| HCT (%) | M | 40–55 | 36–47 | 35.6–55.2 | 33–48.1 | 32.3–45.5 | 37–49 | 36.1–49.7 |
| | F | 38–47 | 35–45 | 36–47 | 30.8–44.7 | 28.1–42.4 | 36–46 | 34.6–46.7 |
| MCV(fl) | M | 76–94 | 78–95 | 75–93 | 63.2–91 | 65–89.5 | 78–98 | 70.1–93.2 |
| | F | 80–98 | 78–95 | 74.5–91 | 62.2–94.5 | 67.4–89.9 | 80–100 | 68.7–96.9 |
| MCH(pg) | M | | 26–32 | 25–31 | - | - | - | 22.5–30.9 |
| | F | | 26–32 | 25–30.8 | | | | 22.1–32 |
| | C | 24–31 | | | | | | |
| MCHC(g/dl) | M | 30.1–35.8 | 32–36 | 32–36 | - | - | - | 30.3–36.1 |
| | F | 30.4–34.2 | 32–36 | 32–35 | | | | 29.8–35.8 |
| PLT(10$^9$/L) | M | 138–364 | 140–385 | 158.5–470 | 119–458 | 110–327 | 150–400 | 186–415 |
| | F | 151–462 | | 198–460.4 | 107–482 | 124–353 | 150–400 | 214–476 |
| Neutrophil(10$^9$/L) | M | | | 1.3–7.4 | | | | 1.03–3.9 |
| | F | | | 01-Jul | | | | 1.13–5.7 |
| | C | 0.9–6.8 | 02-Jul | | 0.9–4.6 | 0.9–3.5 | 1.5–6 | |
| Mixed value(10$^9$/L) | M | 0.3–3.8 | | | 0.2–2.5 | 0.5–3.6 | 0.6–1.5 | 0.41.5 |
| | F | 0.2–1.6 | 0.24–1.6 | - | | | | 0.3–1.5 |
| Lymphocyte(10$^9$/L) | M | 1.2–3.3 | 01-Mar | - | | | | 1.4–3.9 |
| | F | 1.1–3.7 | | | 1.4–4.2 | 1.7–4.7 | 1.5–4.5 | 1.4–3.9 |
| Neutrophil (%) | M | 23.7–64.7 | 40–80 | - | - | - | - | 23.3–58.5 |
| | F | 24.78–71 | | | | | | 24–61.2 |
| **Mixed value (%)** | M | 7.2–36.8 | Apr-18 | - | - | - | - | 5.4–67.3 |
| | F | 4.8–22.7 | | | | | | 5–73.4 |
| **Lymphocyte (%)** | M | | 20–40 | - | - | - | - | 27.7–62.6 |
| | F | | | | | | | 28.4–65 |
| | C | 19.8–61.5 | | | | | | |
| **MPV(fl)** | M | | 7.2–10.4 | - | - | - | - | 8.6–12.3 |
| | F | | 7.5–11.5 | | | | | 8.4–11.2 |
| | C | 8.8–13.1 | | | | | | |

**WBC**: White Blood Cell; **RBC**: Red Blood Cell; **Hgb**: Hemoglobin; **Hct**: Hematocrit; **MCV**: Mean Corpuscular Volume; **MCH**: Mean Corpuscular Hemoglobin; **MCHC**: Mean Corpuscular Hemoglobin Concentration; **PLT**: Platelet; **MPV**: Platelet Mean Volume **M**: Male; **F**: Female; **C**: Combined **N**: Number of participants. **Mixed values**: are values which includes Basophiles, Monocytes & Eosinophiles displaying as one value as the machine is three differential.

Finally, our study was compared with others studies. As shown in Table 3, variations among the difference parameters were noted. For example, the white cell count of the study participants as well as the values from other Africans was lower compared to the instrument or American/European values. RBC parameters were higher in the lower range than southwest Ethiopia. Some inconsistencies were also noted in Platelet counts.

## Discussion

Reference intervals are essential for decision making in clinical diagnosis, to initiate and monitor therapeutic actions, or to provide accurate data for epidemiological purposes. Several factors including age, gender, race, environment, socio-economic conditions, dietary pattern influence laboratory parameters. RIs also depend on the type of instrument, reagents and methods used. Hematological parameters tested in this study were WBC and its differential, RBC with indices and platelets counts, analyzed using the automated Sysmex KX 21-N 3 diff hematology analyzer. Most of the parameters showed a significant difference among gender, which is similar to studies done in Western Kenya [11], Tanzania [18], Southwest Ethiopia [19], African American [22], Kuwaiti [23] and Port Harcourt Nigeria [24]. The RBC, Hgb and HCT were significantly higher in males than females (p< 0.001). The reason for the differences among gender may be due to hormonal variations among males and females. For example, erythropoietin release is different in response to the hormonal production among males and females [25] and also progressive maturation increase in muscle mass resulting in increased RBCs production in males but females having lower level may be due to menstrual blood loss [26].

In contrast in our study the mean value of RBCs and hemoglobin results were significantly different from mean values of studies reported in Kuwaiti [23] and African American [22] among children. The differences may be due to ethnicity [22], altitude variation since Kuwaiti have an average mean altitude of 108m above sea level and nutritional differences [23].

There is also a clinical difference among gender for WBC RI in our study which is higher in females than males similar with studies done in Kuwaiti [23], African American [22], Zimbabwe [21], and western Kenya [27]. This gender difference in the total WBC counts is almost in all ethnic groups. This difference may be due to a genuine biological difference [28], and also due to body composition difference between men and women because women have proportionally more fat mass than men [5, 29]. So, fatty mass is the critical component that influences leukocyte counts [30]. However, Kenyan study did not find significant differences between males and females [11].

When comparing WBC results with similar studies, which are done in African American [22], Kuwaiti [23], Zimbabwe [21] results were higher than our study population. The reason may be as the peoples living in Sub-Saharan countries are exposed to chronic viral and bacterial infections [7] lowers their white blood cells.

The current finding also showed that the mean, median and 95th percentile reference intervals for the platelet counts varies significantly between females and males (p<0.05). This study indicated that the platelet value is higher in females than males similar to other studies done in Kuwaiti [23], Zimbabwe [21], Port Harcourt Nigeria [24] and Southwest Ethiopia [19]. The gender difference in the platelets count is in all ethnic groups. It may be due to biological difference [28], due to proteomic variability as males have higher cytokine and growth factor levels in platelet rich plasma when compared with females [31]. Additionally, due to the differences on serum estrogen level in females may play a role in the increasement of platelet counts. As many studies have revealed that estrogen favorably benefits platelet production [31, 32]. Thrombopoietin also increases in response to regular menstruation [33, 34], due to this the total body iron storage is generally lower in women. Therefore, iron depletion is a well-known factor to stimulate platelet production [35, 36]. However, the Kenyan study does not find a significant differences between males and females [11].

The platelet counts are also lower in our study population than similar studies done in Southwest Ethiopia [19], and Zimbabwean adolescents [21].The cause of platelet count differences among different study areas is unknown [24].

Though limited studies are available on children and adolescents, the neutrophil range in the current study which is lower than the currently in use company derived ranges, is consistent with the findings from Zimbabwean adolescents [21]. In addition, the upper limit of the newly established lymphocyte RI is higher than the company derived value but again consistent with values generated from Zimbabwean adolescents [21]. Comparing to the reference interval generated in southwest Ethiopia from a total of 334 children, the WBC count of both lower and upper limit was higher than the current study especially in males. On the other hand, both the lower and upper limit of RBC count was slightly higher in the current study whereas slightly higher upper limit for Hgb was recorded in male children of the Southeast Ethiopian study. Similarly higher lower & upper limit for PLT count in males and higher lower limit only in Females was recorded as compared to the current study [17].

Taken together, when comparing our values with the currently available RI almost all tests have a variation. In our study WBC results were lower than the currently available RI the reason may be, as we know Africans are more exposed to chronic infections [7]. But the RBC and RBC indices are higher than the currently in use RI, the reason for the increment may be due to the temperate climatic conditions of the study area.

Finally the study checked how much misclassification could have happened as a result of utilizing company derived RIs, which is widely being practiced in our country as in most resource limited settings. The reported result showed such misclassification is noted in most of the hematological parameters by the RI established by manufacturer of the instrument Sysmex KX-21N as compared to the RI established by this study. This may be due to nutritional difference, climate, parasitic and viral infections, ethnic background differences between the current study population and the population used for the company derived values, mostly Caucasians. So, it is better to use the locally established RI for the local populations.

## Conclusion

The hematological parameters for apparently healthy adolescents in this study almost differ from other African countries and western countries. There was a significant difference in Red Blood Cells, MCHC, Hemoglobin, Hematocrit, White Blood Cells and Platelets. All reported results were higher in males than females except platelet and WBC counts were higher in females. The established reference interval in this study will help to diagnosis, treat, and follow up of the clients in comparing to the locally healthy adolescents.

## Supporting information

**S1 Annexes.**
(DOCX)

**S1 File.**
(SAV)

## Acknowledgments

We would like to thank Addis Ababa University, College of Health Science, and Department of Medical Laboratory Sciences. Our sincerely thanks also goes to the health extension workers who select the study participants according to the selection criteria and also to the study participants who gave the biological samples. We also thank Wukro General Hospital administration and workers for helped us in the analysis of hematological indices. Finally, we would like to thank Tigrai health research institution for all the support.

## Author Contributions

**Conceptualization:** Hagos Haileslasie, Aster Tsegaye, Gebreslassie Gebremariam, Kelali Kaleaye, Fitsum Mardu.

**Data curation:** Hagos Haileslasie, Getachew Belay, Gebre Adhanom.

**Formal analysis:** Hagos Haileslasie, Aster Tsegaye, Getachew Belay, Gebreslassie Gebremariam, Gebre Adhanom, Fitsum Mardu.

**Investigation:** Hagos Haileslasie, Brhane Tesfanchal, Ataklti Gebertsadik.

**Methodology:** Hagos Haileslasie.

**Project administration:** Hagos Haileslasie, Kelali Kaleaye, Ataklti Gebertsadik.

**Resources:** Hagos Haileslasie, Lemlem Legesse.

**Software:** Hagos Haileslasie, Kelali Kaleaye, Fitsum Mardu.

**Supervision:** Hagos Haileslasie, Brhane Tesfanchal, Kelali Kaleaye, Lemlem Legesse, Ataklti Gebertsadik.

**Validation:** Hagos Haileslasie, Gebremedhin Gebremichail.

**Visualization:** Hagos Haileslasie, Gebreyohanes Teklehaymanot, Gebreslassie Gebremariam, Gebremedhin Gebremichail, Kelali Kaleaye, Ataklti Gebertsadik.

**Writing – original draft:** Hagos Haileslasie, Gebreyohanes Teklehaymanot, Gebreslassie Gebremariam, Aderajew Gebrewahd.

**Writing – review & editing:** Hagos Haileslasie, Aster Tsegaye, Aderajew Gebrewahd, Gebrehiwet Tesfay.

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
