## [Decision Letter · Decision Letter 0]

27 Sep 2019

PONE-D-19-22528

Establishment of community based hematological reference intervals among apparently healthy adolescents aged 12-17 years in Mekelle city, Tigrai, northern Ethiopia: a cross sectional study

PLOS ONE

Dear Mr. Haileslasie,

Thank you for submitting your manuscript to PLOS ONE. After careful consideration, we feel that it has merit but does not fully meet PLOS ONE’s publication criteria as it currently stands. 

Specifically both reviewers expressed concern over the quality of writing and significant editing and re-wording is required. Also of concern is the fact that the manuscript refers to Figure 1, yet no figure is provided. In addition, concerns over how many adolescents did not give assent to participate and it is unclear if  ethical clearance for listing religion has been obtained. If you choose to submit a revised manuscript, please address all concerns raised by both reviewers.

We would appreciate receiving your revised manuscript by Nov 11 2019 11:59PM. To enhance the reproducibility of your results, we recommend that if applicable you deposit your laboratory protocols in protocols.io, where a protocol can be assigned its own identifier (DOI) such that it can be cited independently in the future. For instructions see: http://journals.plos.org/plosone/s/submission-guidelines#loc-laboratory-protocols

We look forward to receiving your revised manuscript.

Kind regards,

Colin Johnson, Ph.D.

Academic Editor

PLOS ONE

Journal Requirements:

2. Please upload a copy of Figure 1, to which you refer in your text on page 9. If the figure is no longer to be included as part of the submission please remove all reference to it within the text.

3. We note you have included a table to which you do not refer in the text of your manuscript. Please ensure that you refer to Table 6 in your text; if accepted, production will need this reference to link the reader to the Table.

Reviewers' comments:

Reviewer's Responses to Questions

**Comments to the Author**

1. Is the manuscript technically sound, and do the data support the conclusions?

Reviewer #1: Yes

Reviewer #2: Partly

2. Has the statistical analysis been performed appropriately and rigorously? 

Reviewer #1: Yes

Reviewer #2: No

3. Have the authors made all data underlying the findings in their manuscript fully available?

Reviewer #1: Yes

Reviewer #2: Yes

4. Is the manuscript presented in an intelligible fashion and written in standard English?

Reviewer #1: No

Reviewer #2: No

5. Review Comments to the Author

Reviewer #1: Thank you for the opportunity to review “Establishment of community based hematological reference intervals among apparently healthy adolescents aged 12-17 years in Mekelle city, Tigrai, northern Ethiopia: a cross sectional study.” This is a well-designed study that adds to the literature and describes an important task: generating locally relevant reference intervals for safety labs. It was a little hard to follow at times, but with a review of grammar and spelling, should make a good contribution to the literature.

Major comments:

Please review the manuscript throughout with a fluent English speaker. There are many grammatical errors which make reading the paper challenging, particularly in the results section (see page 10 – very hard to follow). Too many to summarize here.

Please include a STROBE style flow diagram to go through numbers screened, enrolled, and included in analysis, showing the numbers and reasons for excluding volunteers at each step. For more information, see: Vandenbroucke JP, von Elm E, Altman DG, Gøtzsche PC, Mulrow CD, Pocock SJ, et al. (2007) Strengthening the Reporting of Observational Studies in Epidemiology (STROBE): Explanation and Elaboration. PLoS Med 4(10): e297. https://doi.org/10.1371/journal.pmed.0040297

Minor comments:

Median age (abstract): report as XX.X, the extra level of precision (15.31) is unnecessary.

Methods, data analysis: how many outliers did you exclude? Should you be doing this? The 95% RI should control for this. Please make sure this is clarified in your results and STROBE flow diagram

Table 2: please clarify where and why you combined values. I presume this was because they were not significantly different? Is this mentioned in the methods?

Page 12: what are “the company’s reference intervals (Table 3)”?

I recommend ending your introduction with a quick statement about what you propose to present in this paper. The last sentence of the introduction should be something like “Here, we present our efforts to generate reference intervals in adolescents .,..” etc.

Reviewer #2: Haileslaslie et al: Hematological reference intervals among adolescents

A prospective community-based study defining hematological reference intervals in apparently healthy adolescents in Ethiopia. A major strength is that participants were randomly selected from the community in the area studied and would be suitable to use for the local lab, at least in the short run. However, the presentation of data is complex and difficult to follow, and it is difficult to assess if these figures are generalizable to similar populations.

I assume that presently RIs from the manufacturer is used. Company values are stated in several tables. Where these figures for adults, or for adolescents, or both? This is needed to evaluate how these new RIs compare to the presently RIs used.

Key words: suggest that standardized expressions are used, e.g. not CBC.

Abstract: could be shortened, omit spelling out EDTAs chemical name, what statistical software was used. Also, numbers for RIs could be omitted, but hematology parameters measured should be included, along with a brief description of gender differences.

Background, page 3, last paragraph. I assume studies on pharmaceuticals are meant when stating “toxicity grading scales”, but these sentences are unclear.

Methods: 3.1 could be shortened, e.g. “Mekelle was formerly the capital”, “Mekelle is considered a special zone”. Weyna-Dega climate and altitude is relevant, but preferably state something about this climate instead of using a local term.

3.2 CLSI guidelines should be accompanied by a reference.

3.3 incl table could be shortened.

3.4 How many adolescents did not give assent to participate? Such figures are needed to know how representative this population was when using this random enrollment design.

3.6 Samples were transported ice-cooled within “allowable time”? What was the maximal time allowed until analysis? And during how long period was the collection of samples performed? Hematology systems sometimes drift over time. Where there any internal or external controls used to assure stability over time for the measurements, and comparability to other labs?

3.8 CLSI suggest Harris Boyd (or Lahti) statistics for partitioning. In the study Mann Whitney U-test was employed. What considerations was behind that decision?

Ethical considerations. Some countries do not allow recording of religious belief in scientific studies, as could be seen in Table 2. Was religion covered by the ethical clearance?

Operational definitions could be omitted, and the information in that section integrated into the Methods section.

Results: Figure 1 shows age, however I cannot find any fig 1.

The 1st paragraph relating figures for numerous RIs could be shortened, figures could be omitted or substantially shortened since they also appear in Table 3. Now table 3 shows combined only for those three parameters where the Mann Whitney U-test turned out with insignificant differences between genders (MCH, Neutrophils, lymphocytes in %). However, even though lymphocyte% is insignificant (0.48), lymphocyte in absolute numbers is highly significant (0.004). Thus, I suggest the authors show all data for males and females as well as combined.

Page 12 starts with “Comparing the upper and lower limits….”. I assume the authors refer to Table 4, not Table 3?

Table 5 is a comparison of RIs from different sources. The United States/Europe lack reference/s. Also, information about ages is relevant for these studies, e.g. in the Ethiopian stud (ref 20) children from the age of 5 years are compared to the present study. This is a major limitation in the presentation, comparisons to gender and age in other studies, and to RIs presently used are difficult to follow. Thius, the practical use even in the area where the study was conducted is unclear.

6. PLOS authors have the option to publish the peer review history of their article (what does this mean?). If published, this will include your full peer review and any attached files.

Reviewer #1: No

Reviewer #2: No

---

## [Author Response · Author response to Decision Letter 0]

4 Dec 2019

Point by point response for reviewers 

Response for reviewer 1

Comment 1: Please review the manuscript throughout with a fluent English speaker.

Response: The grammatical and writing errors are corrected. The manuscript was drafted and rewrites again. 

Comment 2: Median age (abstract): report as XX.X, the extra level of precision (15.31) is unnecessary

Response: The decimal number are corrected in the manuscript and rewritten again

Comment 3: How many outliers did you exclude? Should you be doing this? 

Response: Yes, 20(5.8%) of the participants were excluded due to its outlierness. Please, see on page 9 on the result section.

Comment 4: Please clarify where and why you combined values. I presume this was because they were not

 Significantly different? Is this mentioned in the methods?

Response: No, We included in the new revised manuscript in the result part page 10.

Comment 5: what are “the company’s reference intervals (Table 3)”?

Response: The value which is inserted on the machine using the values of the test leaflet of the specific machine. When a new machine come to install in a Laboratory the machine comes with its own specific leaflets. Inserted the values to the machine to use as a reference 

Comment 6: I recommend ending your introduction with a quick statement about what you propose to present in this paper

Response: We accept your comment and the manuscript is modified. You can see at the last of the introduction part.

Response for reviewer 2

Comment 1: I assume that presently RIs from the manufacturer is used. Company values are stated in several tables. Where these figures for adults, or for adolescents, or both? This is needed to evaluate how these new RIs compare to the presently RIs used.

Response: The inserted values on the CBC machine were for both adolescents and adults. There was no separated value on the machine but I compare the result with the leaflet which I have got separate values for adolescents and adults in the manual CBC machine.

Comment 2: Abstract: could be shortened, omit spelling out EDTAs chemical name, what statistical software was used. Also, numbers for RIs could be omitted, but hematology parameters measured should be included, along with a brief description of gender differences.

Response: We abbreviate the long terms of EDTA and numbers out of the hematological parameters. The data were entered and analyzed by SPSS

version 23 statistical software.

Comment 3: Background, page 3, last paragraph. I assume studies on pharmaceuticals are meant when stating “toxicity grading scales”, but these sentences are unclear.

Response: During the treatment of hematological cases for the patients it is based on the WHO guideline which prepared based on the developed countries. So, it is better to follow clinical cases of the patients in relation to the local population.

Comment 4: Methods: 3.1 could be shortened, e.g. “Mekelle was formerly the capital”, “Mekelle is considered a special zone”. Weyna-Dega climate and altitude is relevant, but preferably state something about this climate instead of using a local term

Response: We tried to shortened and local terms are avoided 

Comment 5: CLSI guidelines should be accompanied by a reference.

 3.3 including table could be shortened.

Response: Tables are slightly modified and CLSI reference is putting on the fifth line of the 3.2 sample size determination part [2, 4].

Comment 6: 3.4 How many adolescents did not give assent to participate? Such figures are needed to know how representative this population was when using this random enrollment design

Response: All of the participants who were present at study area during data collection time and who fulfilled the criteria were voluntarily participated after oriented the aim of the research.

Comment 7: 3.6 Samples were transported ice-cooled within “allowable time”? What was the maximal time allowed until analysis?

Response: The maximum allowable time was 5 hours. I have corrected on the manuscript part

Comment 8: During how long period was the collection of samples performed? Hematology systems sometimes drift over time. 

Response: The data collection was performed from the middle of December to the last of April within 4 months. 

Comment 9: Where there any internal or external controls used to assure stability over time for the measurements, and comparability to other labs?

Response: Yes, We have used both internal and external controls. Comparability was done between Ayder referral Hospital hematology CBC value and Wukro General Hospital. The result was similar related to each other by transporting the sample within 5 hours like that of participant’s sample.

Comment 10: 3.8 CLSI suggests Harris Boyd (or Lahti) statistics for partitioning. In the study Mann Whitney U-test was employed. What considerations were behind that decision?

Response: Harris Boyd (or Lahti) is important for partitioning of the study participants but Mann Whitney U-test is used for analysis of the data to compare the gender difference between males and females based on their median values.

Comment 11: Ethical considerations. Some countries do not allow recording of religious belief in scientific studies, as could be seen in Table 2. Was religion covered by the ethical clearance?

Response: No, because in Ethiopia many research are done without religion ethical clearance. There is no problem participating a religion belief in scientific studies.

Comment 12: Operational definitions could be omitted, and the information in that section integrated into the Methods section.

Response: we have omitted and merge to the methodology parts

Comment 13: Results: Figure 1 shows age, however I cannot find any fig 1.The 1st paragraph relating figures for numerous RIs could be shortened; figures could be omitted or substantially shortened since they also appear in Table 3. Now table 3 shows combined only for those three parameters where the Mann Whitney U-test turned out with insignificant differences between genders (MCH, Neutrophils, lymphocytes in %). However, even though lymphocyte% is insignificant (0.48), lymphocyte in absolute numbers is highly significant (0.004). Thus, I suggest the authors show all data for males and females as well as combined.

Page 12 starts with “Comparing the upper and lower limits….” I assume the authors refer to 

Table 4 not to Table3

 Response: Figure and Tables rearranged and corrected the problem was due to typical errors.

Comment 14: Table 5 is a comparison of RIs from different sources. The United States/Europe lack reference/s Also, information about ages is relevant for these studies, e.g. in the Ethiopian children from the age of 5 years are compared to the present study. This is a major limitation in the presentation; comparisons to gender and age in other studies, and to RIs presently used are difficult to follow. Thus, the practical use even in the area where the study was conducted is unclear.

Response: We put the reference for United States/Europe on the appropriate space on Table5.

---

## [Decision Letter · Decision Letter 1]

27 Jan 2020

PONE-D-19-22528R1

Establishment of community based hematological reference intervals among apparently healthy adolescents aged 12-17 years in Mekelle city, Tigrai, northern Ethiopia: a cross sectional study

PLOS ONE

Dear Mr. Haileslasie,

Thank you for submitting your manuscript to PLOS ONE. After careful consideration, we feel that it has merit but does not fully meet PLOS ONE’s publication criteria as it currently stands. Therefore, we invite you to submit a revised version of the manuscript that addresses the points raised during the review process.

Specifically, the Reviewer thought that the manuscript is of value to the research community, and that if their concerns are addressed, the manuscript should be published. Thus, I have decided a minor revision of the manuscript should be made.Specifically, the comments by the reviewer should be addressed, including the need to incorporate a laboratory analysis section for hemoparasite examination.

We would appreciate receiving your revised manuscript by Mar 12 2020 11:59PM. To enhance the reproducibility of your results, we recommend that if applicable you deposit your laboratory protocols in protocols.io, where a protocol can be assigned its own identifier (DOI) such that it can be cited independently in the future. For instructions see: http://journals.plos.org/plosone/s/submission-guidelines#loc-laboratory-protocols

We look forward to receiving your revised manuscript.

Kind regards,

Colin Johnson, Ph.D.

Academic Editor

PLOS ONE

Reviewers' comments:

Reviewer's Responses to Questions

**Comments to the Author**

1. If the authors have adequately addressed your comments raised in a previous round of review and you feel that this manuscript is now acceptable for publication, you may indicate that here to bypass the “Comments to the Author” section, enter your conflict of interest statement in the “Confidential to Editor” section, and submit your "Accept" recommendation.

Reviewer #3: (No Response)

2. Is the manuscript technically sound, and do the data support the conclusions?

Reviewer #3: Yes

3. Has the statistical analysis been performed appropriately and rigorously? 

Reviewer #3: Yes

4. Have the authors made all data underlying the findings in their manuscript fully available?

Reviewer #3: Yes

5. Is the manuscript presented in an intelligible fashion and written in standard English?

Reviewer #3: Yes

6. Review Comments to the Author

Reviewer #3: The manuscript covers a topic of scientific significance as it generally described the problem and need for establishing reference intervals for the populations of interest.

However, the following observations need some clarifications.

1. I suggest the title to be adjusted as “Community based hematological reference intervals among apparently healthy adolescents aged 12-17 years in Mekelle city, Tigrai, Northern Ethiopia: a cross-sectional study”

2. Topographical errors throughout the manuscript: spelling, capitalization of proper nouns including author names, sentence form (present Vs past) and grammatic problems

3. There are sentences that start with numbers here and there. It is not recommended to start a sentence with numbers please correct.

4. Information which are not related to the manuscript should be removed e.g. in the background section paragraph 2 last 3 sentences have low value to the manuscript.

“Growth is an extremely complex and non-linear biological process, driven by hormonal mechanisms, characterized by an intrinsic variability reflecting environmental, genetic influences and individual adaptive responses. Growth charts are very helpful as reference tools for pediatricians, in order to monitor individual growth and provide therapeutic interventions. Most female’s puberty starts around 12 years but males from 14 years [6].”

5. Repetitions: e.g. last two sentences of Paragraph 4 of Background: please merge them

Adult Africans have been shown to have lower levels of hemoglobin, red blood cells, platelets, and neutrophil counts compared to adults in the developed countries [12-15]. Further, most blood specimens of Africans have been found to have lower Hemoglobin (Hgb), Hematocrit (Hct), Red blood cell count (RBC), mean corpuscular volume (MCV), neutrophil and platelet counts [16].

6. “Sex” should be replaced by “Gender”

7. “Sampling technique” should be “Reference population selection technique”

8. “K” should be specified: what is the value of “K” it should be mentioned

9. Words such as “doesn’t” should be written in full term as “did not”

10. Table 1 is not cited

11. Were there significant differences in RIs of the three kebeles? And also, please clarify kebele? What do you mean by Kebele?

12. Blood sample for hemoparasites, Stool specimens for intestinal parasites examination and urine samples for urinalysis were collected but there is no lab analysis section. There must be laboratory analysis section for hemoparasite examination, Stool examination and urinalysis. A more complete description of the laboratory methods, especially metrological traceability, is required.

13. How do you exclude the influence of infectious diseases such as HIV, Hepatitis on RIS?

14. How representative is the population for their values to be used for reference intervals?

15. Quality control: What was the validation status of the equipment and what QA protocols (maintenance, reproducibility, accuracy, between run, within run, etc) used to validate the analyser used? These establish how accurate the analysers used produce results. How long did samples stay before analysis and how were they kept in case samples were not analyzed immediately?

16. A comment should be made on the pre-analytical factors. Especially given the dependence of cell counts on haemoconcentration, it is necessary to indicate whether subjects are seated or lying, how long a tourniquet was applied for and also how long between collection and analysis.

17. Data analysis: The statement “Data lower than first quartile (Q1-1.5 × IQR), or higher than third quartile (Q3+1.5 × IQR) (Whisker and blot method) were considered as outliers and the outlier was excluded.” Needs reference

18. Data analysis: Do you check normality of data? Was the data normally distributed?

19. Results: Socio demographic characteristics: The use of mean and SD to describe the age distribution is inappropriate as the population age is clearly not Gaussian. I would recommend median, range and interquartile ranges or similar.

20. Result: The numbers in the statement “(15) were outliers (20) and presence of hemolysis (10”) are not clear

21. Table 2 should be deleted. It has no value.

22. Mean and 95% CI should be deleted from table 3 because it is inappropriate as the parameters value are clearly not Gaussian. I recommend the use of Median and Interquartile range.

23. Table 5: should also include other studies, and also check reference 22, is it for US/Europe or Tanzanian?

24. Result and Discussion: Please remove comparisons that made with adult population. Some studies used for comparison are adult population RIs.

25. Discussion: Reference 22 and 23 are about RIs but how they could be a reference for the last statements in the first paragraph of the background?

26. Discussion: paragraph 4 and 5: The references should be revised…..

Overall Recommendation

The manuscript is potentially acceptable for publication if the investigators were able to exhaustively address the suggested changes above.

7. PLOS authors have the option to publish the peer review history of their article (what does this mean?). If published, this will include your full peer review and any attached files.

Reviewer #3: Yes: Bamlaku Enawgaw

---

## [Author Response · Author response to Decision Letter 1]

4 Mar 2020

Comment 1: I suggest the title to be adjusted as “Community based hematological reference intervals among apparently healthy adolescents aged 12-17 years in Mekelle city, Tigrai, Northern Ethiopia: a cross-sectional study”

Response 1: We have changed the Title to “Community based hematological reference intervals among apparently healthy adolescents aged 12-17 years in Mekelle city, Tigrai, Northern Ethiopia: a cross-sectional study”

Comment 2: Topographical errors throughout the manuscript: spelling, capitalization of proper nouns including author names, sentence form (present Vs past) and grammatical problems

Response 2: We have corrected the topographical errors

Comment 3: There are sentences that start with numbers here and there. It is not recommended to start a sentence with numbers please correct.

Response 3: We have corrected it (Example: Summary Result part) 

 Comment 4: Information which is not related to the manuscript should be removed e.g. in the background section paragraph 2 last 3 sentences have low value to the manuscript.

“Growth is an extremely complex and non-linear biological process, driven by hormonal mechanisms, characterized by an intrinsic variability reflecting environmental, genetic influences and individual adaptive responses. Growth charts are very helpful as reference tools for pediatricians, in order to monitor individual growth and provide therapeutic interventions. Most female’s puberty starts around 12 years but males from 14 years [6].”

Response 4: We have tried to remove the unwanted part and tried to rearrange it.

Comment 5: Repetitions: e.g. last two sentences of Paragraph 4 of Background: please merge them. Adult Africans have been shown to have lower levels of hemoglobin, red blood cells, platelets, and neutrophil counts compared to adults in the developed countries [12-15]. Further, most blood specimens of Africans have been found to have lower Hemoglobin (Hgb), Hematocrit (Hct), Red blood cell count (RBC), mean corpuscular volume (MCV), neutrophil and platelet counts [16].

Response 5: The unwanted repetition tried to merged and rearranged on Paragraph 4 the Background part

Comment 6: “Sex” should be replaced by “Gender”

Response 6: We tried to replace Sex by Gender (Summary background part, background section paragraph 2, discussion parts and also in other parts)

Comment 7: “Sampling technique” should be “Reference population selection technique”

Response 7: We have tried to replace “Sampling technique” by “Reference population selection technique”

Comment 8: “K” should be specified: what is the value of “K” it should be mentioned

Response 8: “K” is the interval of data taking households from the first household to the next (second) household. Kth=total number of households in the 3 sub cities divided by total number of participants. Kth=199 the participants data were collected every 199th household.

Comment 9: Words such as “doesn’t” should be written in full term as “did not”

Response 9: We tried to corrected such like terms (discussion Paragraph 3)

Comment 10: Table 1 is not cited

Response 10: The data was obtained from the municipality of Mekelle city (unpublished data). 

Comment 11: Were there significant differences in RIs of the three kebeles? And also, please clarify kebele? What do you mean by Kebele?

Response 11: There was no significance difference in RIs among the three kebeles. Kebele is part of a sub city which is a small administrating part of the government.

Comment 12: Blood sample for hemoparasites, Stool specimens for intestinal parasites examination and urine samples for urinalysis were collected but there is no lab analysis section. There must be laboratory analysis section for hemoparasite examination, Stool examination and urinalysis. A more complete description of the laboratory methods, especially metrological traceability, is required.

Response 12: We have tried to incorporate the Lab analysis section

Comment 13: How do you exclude the influence of infectious diseases such as HIV, Hepatitis on RIS?

Response 13: The health extension professionals have a family folder which follows the health status continuously. So, by following stringent criteria for exclusion before sample collection and those apparently healthy individuals sample was screened in the Tigrai blood bank using an ELISA technique. 

Comment 14: How representative is the population for their values to be used for reference intervals?

Response 14: The participants were 100% representative for the city because of their systematically selection methods. The three sub cities were selected randomly from the total of seven sub cities. According to CLSI guideline in order to say representative it is better to cover more than 30% of the total subjects.

Comment 15: Quality control: What was the validation status of the equipment and what QA protocols (maintenance, reproducibility, accuracy, between run, within run, etc) used to validate the analyzer used? These establish how accurate the analyzers’ used produce results. How long did samples stay before analysis and how were they kept in case samples were not analyzed immediately?

Response 15: The Hospital was on the process of accreditation. Its maintenances protocol was controlled according to its standard procedures continuously. Accuracy was done by using the daily QC tests (Low, Normal and High values) and also reproducibility was checked. Between run and within run also checks to avoid carry over between former and later samples. The sample was preserved within 2-8oc cold chain box until analysis. The CBC sample was done with in five hours of collection. The 2-8oc preserved sample was put in the room temperature for 30 minutes before analysis.

Comment 16: A comment should be made on the pre-analytical factors. Especially given the dependence of cell counts on haemoconcentration, it is necessary to indicate whether subjects are seated or lying, how long a tourniquet was applied for and also how long between collection and analysis.

Response 16: During the blood collection time participants was seated on a comfortably chair before collection and tourniquet was applied for less than one minute until the vein is visible and anchored with the syringe. One’s the vein is visible and anchored with the syringe the tourniquet was removed. The collected sample was analyzed with in five hours. 

Comment 17: Data analysis: The statement “Data lower than first quartile (Q1-1.5 × IQR), or higher than third quartile (Q3+1.5 × IQR) (Whisker and blot method) were considered as outliers and the outlier was excluded.” Needs reference

Response 17: We have putting the reference for the outliers on the data analysis part

 Comment 18: Data analysis: Do you check normality of data? Was the data normally distributed?

Response 18: Yes, We have checked the normality before doing the analysis. But, the data was not normally distributed. 

Comment 19: Results: Socio demographic characteristics: The use of mean and SD to describe the age distribution is inappropriate as the population age is clearly not Gaussian. I would recommend median, range and interquartile ranges or similar.

Response 19: We take the comments and corrected it.

Comment 20: Result: The numbers in the statement “(15) were outliers (20) and presences of hemolysis (10”) are not clear

Response 20: We have tried to make clear and easily understandable

Comment 21: Table 2 should be deleted. It has no value.

Response 21: Table 2 is removed completely

Comment 22: Mean and 95% CI should be deleted from table 3 because it is inappropriate as the parameters value are clearly not Gaussian. I recommend the use of Median and Interquartile range.

Response 22: Ok, Mean is removed from the Table but 95% CI is important for RIs than the Inter Quartile range 

Comment 23: Table 5: should also include other studies, and also check reference 22, is it for US/Europe or Tanzanian?

Response 23: It is for Tanzanian it is a typical error we have corrected it 

Comment 24: Result and Discussion: Please remove comparisons that made with adult population. Some studies used for comparison are adult population RIs.

Response 24: We have tried to reject the comparison with adults 

 Comment 25: Discussion: Reference 22 and 23 are about RIs but how they could be a reference for the last statements in the first paragraph of the background?

Response 25: We have replaced and rearranged the unnecessary references 

Comment 26: Discussion: paragraph 4 and 5: The references should be revised…..

Response 26: Discussion: paragraph 4 is tried to rearranged with other references

---

## [Decision Letter · Decision Letter 2]

17 Mar 2020

PONE-D-19-22528R2

Community based hematological reference intervals among apparently healthy adolescents aged 12-17 years in Mekelle city, Tigrai, northern Ethiopia: a cross sectional study

PLOS ONE

Dear Mr. Haileslasie,

Thank you for submitting your manuscript to PLOS ONE. After careful consideration, we feel that the manuscript is much improved and invite you to submit a revised version of the manuscript that addresses the points raised during the review process.

Specifically, while the response to reviewers were acceptable, reviewers commented that incorporation of the rebuttal information into the document, especially  the methods section, would improve the manuscript.

We would appreciate receiving your revised manuscript by May 01 2020 11:59PM. To enhance the reproducibility of your results, we recommend that if applicable you deposit your laboratory protocols in protocols.io, where a protocol can be assigned its own identifier (DOI) such that it can be cited independently in the future. For instructions see: http://journals.plos.org/plosone/s/submission-guidelines#loc-laboratory-protocols

We look forward to receiving your revised manuscript.

Kind regards,

Colin Johnson, Ph.D.

Academic Editor

PLOS ONE

Reviewers' comments:

Reviewer's Responses to Questions

**Comments to the Author**

1. If the authors have adequately addressed your comments raised in a previous round of review and you feel that this manuscript is now acceptable for publication, you may indicate that here to bypass the “Comments to the Author” section, enter your conflict of interest statement in the “Confidential to Editor” section, and submit your "Accept" recommendation.

Reviewer #3: (No Response)

2. Is the manuscript technically sound, and do the data support the conclusions?

Reviewer #3: Yes

3. Has the statistical analysis been performed appropriately and rigorously? 

Reviewer #3: Yes

4. Have the authors made all data underlying the findings in their manuscript fully available?

Reviewer #3: Yes

5. Is the manuscript presented in an intelligible fashion and written in standard English?

Reviewer #3: Yes

6. Review Comments to the Author

Reviewer #3: My previous comments are addressed in the line by line response but some responses especially responses for my comments in the method section should be included in the main manuscript. Before publication the following minor comments should be addressed.

1. Table 1 is not cited in the text

2. My previous comment in method section is addressed but it is not included in the manuscript. The responses given for my comments should be included in the manuscript. (Comments 11, 13, 15, 16, 18)

3. Table needs modification. The RI is not clearly indicated. The table column is better to be rearranged as:

Parameter

Gender

N

Median

RI (95th percentile)

2.5th Percentile 90% CI

2.5th Percentile 90% CI

P value

4. Table 2: Partition in gender is not needed for parameter which have no significant difference between male and female. (MCH. Neutrophil, Lymphocyte and MPV)

5. Table 2: “mixed value” needs definition.

6. Discussion: As indicated in the result section, most of the parameters show a statistically significant difference between male and Female. There should be a comment on statistically Vs Clinically significant difference.

7. Mean should be removed

8. Sex should be replaced by gender

7. PLOS authors have the option to publish the peer review history of their article (what does this mean?). If published, this will include your full peer review and any attached files.

Reviewer #3: Yes: Bamlaku Enawgaw

---

## [Author Response · Author response to Decision Letter 2]

28 Apr 2020

Especially reviewer 3 asses the manuscript in short period of time in detail we appreciate him

---

## [Editor Report · Decision Letter 3]

29 Apr 2020

PONE-D-19-22528R3

Community based hematological reference intervals among apparently healthy adolescents aged 12-17 years in Mekelle city, Tigrai, northern Ethiopia: a cross sectional study.

PLOS ONE

Dear Mr. Haileslasie,

Thank you for submitting your manuscript to PLOS ONE. After careful review, we feel that the revised manuscript is significantly improved, but should include the responses you provided the referee in the method section of the main manuscript. Therefore, we invite you to submit a revised version of the manuscript that addresses the points raised during the review process.

We would appreciate receiving your revised manuscript by Jun 13 2020 11:59PM. To enhance the reproducibility of your results, we recommend that if applicable you deposit your laboratory protocols in protocols.io, where a protocol can be assigned its own identifier (DOI) such that it can be cited independently in the future. For instructions see: http://journals.plos.org/plosone/s/submission-guidelines#loc-laboratory-protocols

We look forward to receiving your revised manuscript.

Kind regards,

Colin Johnson, Ph.D.

Academic Editor

PLOS ONE

---

## [Author Response · Author response to Decision Letter 3]

18 May 2020

We appreciate for the reviewers for responding the activity in short period of time

---

## [Editor Report · Decision Letter 4]

20 May 2020

Community based hematological reference intervals among apparently healthy adolescents aged 12-17 years in Mekelle city, Tigrai, northern Ethiopia: a cross sectional study.

PONE-D-19-22528R4

Dear Dr. Haileslasie,

We are pleased to inform you that your manuscript has been judged scientifically suitable for publication and will be formally accepted for publication once it complies with all outstanding technical requirements.

With kind regards,

Colin Johnson, Ph.D.

Academic Editor

PLOS ONE
---

## [Editor Report · Acceptance letter]

31 Aug 2020

PONE-D-19-22528R4 

Community based hematological reference intervals among apparently healthy adolescents aged 12-17 years in Mekelle city, Tigrai, northern Ethiopia: a cross sectional study. 

Dear Dr. Haileslasie:

I'm pleased to inform you that your manuscript has been deemed suitable for publication in PLOS ONE. Congratulations! Your manuscript is now with our production department. 

Kind regards, 

on behalf of

Dr. Colin Johnson 

Academic Editor

PLOS ONE